# Characterization of a PERK Kinase Inhibitor with Anti-Myeloma Activity

**DOI:** 10.3390/cancers12102864

**Published:** 2020-10-05

**Authors:** Tina Bagratuni, Dimitrios Patseas, Nefeli Mavrianou-Koutsoukou, Christine Ivy Liacos, Aimilia D. Sklirou, Pantelis Rousakis, Maria Gavriatopoulou, Evangelos Terpos, Ourania E. Tsitsilonis, Ioannis P. Trougakos, Efstathios Kastritis, Meletios A. Dimopoulos

**Affiliations:** 1Department of Clinical Therapeutics, School of Medicine, National and Kapodistrian University of Athens, 11528 Athens, Greece; dimitrispat@biol.uoa.gr (D.P.); nmavrianou@med.uoa.gr (N.M.-K.); liakou@med.uoa.gr (C.I.L.); mgavria@med.uoa.gr (M.G.); eterpos@med.uoa.gr (E.T.); ekastritis@med.uoa.gr (E.K.); mdimop@med.uoa.gr (M.A.D.); 2Department of Cell Biology and Biophysics, Faculty of Biology, National and Kapodistrian University of Athens, 15784 Athens, Greece; asklirou@biol.uoa.gr (A.D.S.); itrougakos@biol.uoa.gr (I.P.T.); 3Department of Animal and Human Physiology, Faculty of Biology, National and Kapodistrian University of Athens, 15784 Athens, Greece; prousakis@biol.uoa.gr (P.R.); rtsitsil@biol.uoa.gr (O.E.T.)

**Keywords:** apoptosis, bortezomib, cell survival, multiple myeloma, PERK, UPR

## Abstract

**Simple Summary:**

Multiple myeloma is a bone marrow cancer that represents a severe health threat. The drugs used nowadays in chemotherapy often encounter resistance leading to a dramatic loss of their efficacy, which consequently affects patients’ survival. Previous studies have shown that the protein kinase R (PKR)-like ER kinase (PERK) pathway, which is one of the three branches of the unfolded protein response, is highly activated in multiple myeloma, possibly contributing to the chemotherapy resistance that these patients develop. In this study, we have used the compound GSK2606414, which is a PERK inhibitor, and found that myeloma cells are highly sensitive to this molecule. These effects were more pronounced when the inhibitor was used in combination with an anti-myeloma drug such as the proteasome inhibitor bortezomib, suggesting that the PERK pathway could be a potential therapeutic target for the treatment of multiple myeloma patients.

**Abstract:**

Due to increased immunoglobulin production and uncontrolled proliferation, multiple myeloma (MM) plasma cells develop a phenotype of deregulated unfolded protein response (UPR). The eIF2-alpha kinase 3 [EIF2αK3, protein kinase R (PKR)-like ER kinase (PERK)], the third known sensor of endoplasmic reticulum (ER) stress, is a serine-threonine kinase and, like the other two UPR-related proteins, i.e., IRE1 and ATF6, it is bound to the ER membrane. MM, like other tumors showing uncontrolled protein secretion, is highly dependent to UPR for survival; thus, inhibition of PERK can be an effective strategy to suppress growth of malignant plasma cells. Here, we have used GSK2606414, an ATP-competitive potent PERK inhibitor, and found significant anti-proliferative and apoptotic effects in a panel of MM cell lines. These effects were accompanied by the downregulation of key components of the PERK pathway as well as of other UPR elements. Consistently, *PERK* gene expression silencing significantly increased cell death in MM cells, highlighting the importance of PERK signaling in MM biology. Moreover, GSK2606414, in combination with the proteasome inhibitor bortezomib, exerted an additive toxic effect in MM cells. Overall, our data suggest that PERK inhibition could represent a novel combinatorial therapeutic approach in MM.

## 1. Introduction

Plasma cells are mainly characterized by the large quantity of immunoglobulins (Igs) that they synthesize and secrete. These Igs are folded into their tertiary structures within the endoplasmic reticulum (ER), where the unfolded protein response (UPR) maintains the equilibrium between the rate of protein production and the capacity for nascent protein folding. Activation of the UPR during proteotoxic stress results in a bias of translation towards the synthesis of chaperone proteins involved in protein folding [1]. Hence, the presence of correctly folded nascent Igs within the ER provides an effective checkpoint for plasma cell survival. Multiple myeloma (MM) plasma cells produce large amounts of Igs and thus they are highly dependent on the UPR regulatory axis. The main function of the UPR activation is to suppress the synthesis of new polypeptides that cannot be correctly processed and/or folded in order to ensure correct folding, processing, export and/or degradation of proteins already synthesized. The UPR is mediated through the activation of three transmembrane stress sensors, namely eukaryotic translation initiation factor 2 alpha kinase 3 [also known as protein kinase R (PKR)-like ER kinase (PERK)], the activating transcription factor 6 (ATF6) and the inositol-requiring enzyme 1 (IRE1) [2]. These proteins are maintained in an inactive state through a physical interaction between their ER lumen domains and the ER chaperone binding immunoglobulin protein (BiP). When unfolded proteins accumulate in the ER, inducing ER stress, the interaction between BiP and its client proteins is destabilized, allowing the dimerization of these transmembrane signaling proteins and their autocatalytic activation [3,4,5,6].

PERK is a BiP-bound type 1 transmembrane protein of around 120 kDa with a C-terminal cytosolic domain that possesses serine/threonine kinase activity and an IRE1-like ER luminal domain [3]. Upon accumulation of unfolded proteins in the ER lumen, PERK dimerization and phosphorylation inactivates the general translation initiation factor EIF2α, which is required for the 80S ribosome assembly, resulting in a general shutdown of polypeptide synthesis [7]. Although phosphorylation of eIF2α inhibits general mRNA translation initiation, it is required for the selective translation of several mRNAs such as those encoding activating transcription factor 4 (ATF4) [8,9,10]. ATF4 is a transcription factor in the cAMP-response element binding (CREB) family and activates many genes involved in controlling the UPR including chaperones such as BiP and GRP94, genes involved in suppressing oxidative stress, as well as genes involved in amino acid metabolism and transport [11]. ATF4 also induces the expression of the CCAAT/enhancer-binding protein (C/EBP) homologous protein (CHOP or GADD153) and the marker of ER stress-induced apoptosis growth arrest DNA damage 34 (GADD34) [12,13,14].

The role of PERK activation in cell proliferation and apoptosis has been extensively studied in various tumors [15,16,17]. PERK has been shown to display a protective role, but it can also induce cell death mechanisms due to prolonged activation, suggesting that further investigation on its specific role is needed. Specifically, it has been found that, in addition to promoting survival, PERK can also suppress tumor growth of advanced carcinomas such as squamous and colorectal carcinoma cells [15]. In other studies, it was found that the integrated stress response coordinated by PERK can both promote and inhibit medulloblastoma tumorigenesis by regulating apoptosis [16,17]. In addition, activation of PERK has been implicated in a wide variety of cancers, as it enhances responses to chemotherapy [18,19,20,21,22,23,24], while knockdown of PERK in MM cells resulted in autophagic cell death [25]. Moreover, in a bortezomib (BTZ)-resistant subpopulation of myeloma cells, it was found that BTZ resistance can be reversed by eIF2α inhibition [26].

Based on the hypothesis that targeting various components of the UPR may offer a therapeutic benefit in protein over-secreting tumors such as MM, we sought to investigate the biological effects of a highly selective inhibitor of PERK as a potential anti-MM agent.

## 2. Results

### 2.1. Human Myeloma Cell Lines and CD138^+^ Myeloma Cells from MM Patients Express High Levels of PERK

We screened a panel of eight human myeloma cell lines (HMCLs) (L363, H929, U266, JJN3, RPMI-8226, OPM2, KMS11 and JIM3) for the expression of *PERK* mRNA (Figure 1A_1_) and protein (Figure 1A_2_,A_3_) levels. According to protein expression analyses, the H929 and L363 cell lines express the highest PERK protein levels, whereas OPM2 and JJN3 showed minimal expression levels.

The expression of *PERK* was also determined in primary CD138^+^ myeloma cells isolated from the bone marrow of 25 patients at the time of diagnosis. *PERK* mRNA was highly expressed in almost all patients. Specifically, almost 75% of MM patients (19 out of 25) expressed high levels of *PERK*, and 30% (7 out of 25) expressed almost 20–80 fold higher *PERK* compared to the ES2 ovarian cancer cell line that was used as control (Figure 1B). In addition, *PERK* mRNA expression in patients seems to be much higher than in the HMCLs, with a mean expression of almost 50 times higher vs. the mean expression of HMCLs. Thus, the abundant expression of *PERK* mRNA in human myeloma cells indicates that UPR signaling through PERK may play an important role in plasma cell biology.

### 2.2. The PERK Specific Inhibitor GSK2606414 Decreases MM Cell Survival and Induces MM Cell Apoptosis

Then, we investigated the effects of GSK2606414, a selective PERK inhibitor, on MM cell viability. HMCLs were incubated for 24, 48 and 72 h with increasing concentrations of GSK2606414. The results showed a progressive decrease in cell viability in a dose- and time-dependent manner in all cell lines examined (Figure 2). Specifically, we found that treatment of cells with 1–100 μM of GSK2606414 results in a dose and time-dependent inhibition of cell viability in the majority of HMCLs studied; the most pronounced effects were seen after 48 h of treatment. Notably, the H929, KMS11, L363 and U266 HMCLs showed the higher sensitivity in PERK inhibition, whereas OPM2, JJN3 and JIM3 cells were more resistant; the IC_50_ (inhibition concentration 50) values per cell line are presented in Table 1. Moreover, there was a significant correlation of higher *PERK* expression levels with sensitivity to PERK inhibition (rho = −0.7719, *p* = 0.009), further supporting the notion that PERK may contribute to MM cell survival (Appendix A).

To evaluate the specific anti-proliferative effects of GSK2606414 in MM cells lines expressing high levels of PERK (i.e., H929, L363, KMS11 and U266), we stained cells with Annexin V/propidium iodide (PI) to measure the apoptotic and necrotic cell populations. Treatment of these HMCLs with the respective IC_50_ doses of GSK2606414 for 48 h resulted in a 7–30% increase in the early apoptotic population and a 1–46.5% increase in the late apoptotic/necrotic populations vs. non-treated cells (Figure 3A).

### 2.3. PERK Downregulation Reduces Cell Survival and Induces Apoptosis of MM Cells

In order to phenocopy PERK-inhibition-mediated effects and further confirm the role of PERK in HMCL survival, we performed RNAi-mediated silencing of *PERK* gene expression. These experiments were done in H929 and L363 cells that expressed high endogenous PERK levels and which were highly responsive to GSK2606414-induced PERK inhibition. We found that the knockdown of *PERK* mRNA (Figure 3B_1_) and protein (Figure 3B_2_) expression significantly decreased cell survival in H929 (*p* = 0.003) and L363 cells (*p* = 0.01), as determined by both the WST1 (Figure 3B_3_) and Trypan blue assays (H929 cells, *p* = 0.05; L363 cells, *p* = 0.1) (Figure 3B_4_). Thus, either pharmacologic PERK inhibition or *PERK* genetic downregulation inhibits MM cell growth and induces apoptosis, suggesting that the increased PERK basal expression levels in MM cells are essential for their survival and proliferation.

### 2.4. The GSK2606414 Inhibitor Suppresses the PERK Signaling Axis in MM Cells

As myeloma plasma cells produce a large amount of Igs, integrated stress response mechanisms and UPR are constantly activated and, thus, ATF4 and CHOP expression levels are higher than in normal plasma cells [27]. Hence, we sought to determine the effects of PERK inhibition on PERK activity through the differential expression of downstream targets of PERK. To this end, MM cells expressing the highest mRNA and protein levels of PERK, i.e., H929 and L363, were treated with 10 μM GSK2606414 for 48 h; then, we analyzed the effects of the inhibitor on the downstream signaling modules of the PERK pathway. The transcript expression analysis showed that PERK inhibition decreased the mRNA levels of endogenous *PERK* and *ATF4* in both cell lines, while *CHOP* levels slightly increased in H929 cells and were significantly upregulated in L363 cells (*p* < 0.001) proportionally to PERK reduction (Figure 4A_1_). Immunoblotting analysis showed a slight suppression of ATF4 and a more pronounced suppression of total EIF2α and pEIF2α in both cell lines, whereas CHOP levels remained almost unaltered (Figure 4A_2_,A_3_). We also investigated the effect of PERK inhibition on the activity of other UPR branches, namely ATF6 and XBP1 splicing (XBP1s), which mainly deliver pro-survival signals. We found that PERK inhibition minimally decreased ATF6 and spliced XBP1 protein expression levels compared to non-treated cells. As in the case of the GSK2606414 inhibitor, PERK knockdown in H929 and L363 cells resulted in reduced ATF4 and EIF2α protein levels, as well as of EIF2α phosphorylation; however, ATF6 levels were upregulated in both cell lines, whereas spliced XBP1 and CHOP were decreased only in L363 cells (Figure 4B_1_,B_2_).

This readout indicates that PERK upregulation in MM cells may contribute to apoptosis evasion via CHOP induction; the pro-survival mechanism is also affected via the downregulation of ATF4 and proteins involved in other UPR branches, such as ATF6 and spliced XBP1.

### 2.5. Treatment of MM Cells with GSK2606414 Differentially Regulates UPR-Related Genes

To determine whether GSK2606414 can affect UPR-related gene expression even under conditions of sustained ER stress, H929 cells (which express high endogenous mRNA levels of PERK) were treated with 10 μM GSK2606414 for 24 h and were further subjected to ER stress conditions by treatment with Tunicamycin (TM) for another 24 h. Cells were harvested and analyzed (vs. controls) for changes in RNA expression of 84 UPR-related genes. We observed that transcription levels of 40 genes changed fivefold in GSK2606414/TM-treated cells compared to TM-exposed cells. Thirty of these genes (*ERN1*, *ERN2*, *XBP1*, *DDIT3*, *CEBPB*, *PPP1R15A*, etc.) were downregulated by >fivefold, whereas 10 of these genes (*HERPUD1*, *CREB3L3*, *HSPA2*, *HSPA1B*, etc.) were upregulated similarly (Figure 5). These results show that GSK2606414 affects many other functional UPR components in addition to the PERK-ATF4 pathway, mainly delivering a pro-apoptotic signal through the downregulation of pro-survival genes, such as *XBP1* and *IRE1*.

### 2.6. GSK2606414 Exerts Synergistic to BTZ Effects in MM Cells

Given the anti-MM activity of proteasome inhibitors and their importance in MM treatment in the clinic [28], we sought to evaluate the effects of PERK inhibition in MM cells in combination with BTZ, a selective and potent inhibitor of the 26S proteasome, which also induces ER stress and UPR [29,30,31,32]. For this study, H929 and L363 cells were pre-treated with an IC_50_ dose of GSK2606414 for 24 h and were then subjected to BTZ treatment for another 24 h. The combinatorial treatment of GSK2606414 plus BTZ resulted in an additive apoptotic effect, where the late apoptotic/necrotic population increased by almost 20% in H929 cells compared to only BTZ-treated cells (94.0% vs. 77.2%), while, in the L363 cell line, the early apoptotic population increased by 20% compared to BTZ-only treated cells (55.7% vs. 35.5%) (Figure 6A).

Cell pre-treatment with GSK2606414 also resulted in significantly reduced cell viability in both cell lines when compared to BTZ-only treated cells (H929, *p* = 0.004; L363, *p* = 0.006) (Figure 6B). Transcriptional analysis showed that GSK2606414 combined with BTZ resulted in reduced *ATF4* and *CHOP* gene expression levels by 20% (*p* = 0.014) and 60% (*p* = 0.004), respectively, in H929 cells, and by 20% (*p* = 0.005) and 15% (*p* = 0.022) in *ATF4* and *CHOP* genes, respectively, in L363 cells (Figure 6C_1_). This intervention also consistently suppressed the ATF4 and CHOP protein expression levels (Figure 6C_2_).

### 2.7. GSK2606414 Differentially Regulates Apoptotic Pathways in MM cells

To determine the effects of GSK2606414 on apoptotic pathways in more detail, lysates from H929 and L363 cells before and after treatment with GSK2606414 and/or BTZ were analyzed by using an array detection system in order to determine the expression of apoptotic proteins (Figure 7A_1_, Appendix A). Our results showed that treatment with GSK2606414 alone resulted in the upregulation of more than 20 apoptotic proteins, including TRAIL, TNFRSF6, phospho-RAD, etc. compared to untreated cells (Figure 7A_2_, Appendix A). In addition, treatment with GSK2606414 and BTZ resulted in the upregulation of all 35 apoptosis-related proteins by two- to fivefold compared to BTZ-only treated cells (Figure 7A_3_, Appendix A).

## 3. Discussion

Developing agents that would successfully target components of a signaling pathway associated with the precise pathogenic molecular driver for each individual cancer is the key to effective molecular therapies. These pathways are often part of oncogenic networks whose effective inhibition would sensitize tumor cells and successfully drive them to apoptosis and abrogation of the malignant state. The development of pharmacological tools that selectively modulate a druggable enzyme greatly facilitates the investigation of the protein’s biological function and potential therapeutic application. UPR signaling has been shown to play a fundamental role in the development and regulation of tumors, particularly those with a secretory phenotype such as MM. Several studies using animal models have shown that manipulating various UPR proteins by gain- or loss-of-function genetic interventions can trigger antitumor effects [33,34]. PERK is one of the main primary UPR effectors and the development of PERK inhibitors has been of particular interest, since recent evidence implicates PERK as a contributor to initiation and progression events in cancer as well as a supporter of cancer resistance to chemotherapy [8,35]. Therefore, inhibiting PERK in cancer cells may suppress their ability to adapt in stress conditions, leading to apoptosis and/or tumor growth inhibition. GSK2606414 has been characterized as one of the first-in-class selective PERK inhibitors identified from a kinase inhibitor library. GSK2606414 has been found to be highly potent for inhibiting PERK in vitro with an IC_50_ dose of lower than 1 nM and has also shown the inhibition of tumor growth in a human tumor xenograft in mice [36,37].

Our study reports the anti-myeloma effects of GSK2606414 in human myeloma cells. Initially, our study revealed that PERK is expressed in myeloma cells, as demonstrated by the expression patterns of our panel of myeloma cell lines as well as of CD138^+^ plasma cells isolated from selected myeloma patients’ bone marrow samples. In a previous study, it was shown that knockdown of PERK resulted in an autophagic cell death response, suggesting that PERK activation is a necessary element for the metabolic transformation of a plasma cell to a myeloma cell, but also as an impediment of the apoptotic response, revealing the dual activity of PERK, as both a protector and a cell death promoter in MM [25]. Given the expression profile of PERK in myeloma cells, we showed that treatment with GSK2606414 resulted in a dose-dependent anti-proliferative outcome in eight myeloma cell lines. Interestingly, myeloma cell lines with high expression levels of PERK showed increased sensitivity to the PERK inhibitor, and demonstrated lower IC_50_ values and faster responses, highlighting the on-target effects of the selective inhibitor. Additionally, PERK inhibition prompted high apoptotic effects in the four most highly PERK-expressing myeloma cell lines, with an increased apoptotic signal of 8–30% compared to non-treated cells, in agreement with the results of the WST1 assay. These effects were coupled with the suppression of the main molecular targets of PERK such as ATF4, EIF2α and phospho-EIF2α, but not CHOP.

Interestingly, our results showed that PERK inhibition also slightly affected the pro-survival molecules of the two other branches of the UPR, namely XBP1s and ATF6, suggesting that it might also exert an overall anti-UPR inhibitory effect. This was also shown by the total UPR expression profile and post cell-treatment with GSK2606414, which revealed the suppression of major UPR driver genes, including *IRE1* (*ERN1*) and *XBP1*, which are key players in MM tumor growth [38,39]. In addition, treatment of MM cells with GSK2606414 downregulated the *CEBPB* gene, which regulates transcription factors critical for the proliferation and survival of MM cells [40], and the *PPP1R1A* gene, which is upregulated by ATF4 and provides a negative feedback loop by dephosphorylating eIF2A [41]. On the other hand, MM cell exposure to GSK260614 upregulated genes such as *HSPA2* and *HSPA1B*, which are members of the heat-shock protein family and studies have shown that breast cancer patients overexpressing *HSPA2* exhibited longer survival [42]. In addition, treatment of MM cells with GSK2606414 resulted in the upregulation of *HERPUD1* gene expression, which is an ER stress responsive gene that was previously connected to reduced tumor-associated expression in prostate cancer cells [43]. Across the genes differentially expressed upon treatment with GSK260641, there are several genes that are directly associated with the PERK pathway such as *DDIT3* and *CEBPB* through *ATF4*, while most genes are indirectly connected to the PERK pathway, such as *HEPPUD1*, which is regulated by the ER stress-specific branch of the UPR [44]. As with the IRE1-XBP1 pathway, which is a key pathway in MM biology, we hypothesized that, through the inhibition of one of the three branches of UPR, cells are in a process of adaptation to the changes that might occur by the suppression of PERK-related genes, activating other branches of the UPR by the positive or negative regulation of UPR-related genes. In contrast to PERK inhibition, although *PERK* knockdown suppressed the expression of ATF4 as well as of total and phosphorylated EIF2α, it differentially affected the expression levels of ATF6 and XBP1 in the two MM cell lines assayed, indicating a differential mechanism of action in the supporting branches of the UPR such as IRE1 and ATF6.

The activated UPR in plasma cells could potentially suppress cellular sensitivity to pharmacological compounds by emerging resistance to therapeutic agents. Thus, we investigated whether GSK2606414 exerts synergistic effects with BTZ in MM cells. Our results indeed showed that the combination of GSK2606414 and BTZ led to increased apoptosis and decreased cell survival compared to solely BTZ-treated cells. To further evaluate the apoptosis phenotype seen by the Annexin/PI technique, we screened these samples with an array detection system, which detects 35 human apoptosis-related proteins. Our results showed that the Annexin/PI phenotype was also coupled with the over-expression of many apoptotic proteins when (a) GSK2606414-treated cells were compared with non-treated cells, as well as when (b) GSK2606414/BTZ-treated cells were compared to BTZ-treated cells. Given the dual function of PERK both in cell survival and apoptosis [15,45], we sought to determine whether GGSK2606414 on its own or, more interestingly, in combination with BTZ, would drive cells to increased apoptosis and hence sensitize cells to more effective combination therapies. Among the apoptotic proteins upregulated, survivin has been previously described as being associated with ER stress and UPR gene expression in colonic epithelial cells [46]. Furthermore, claspin has also been previously reported as a negative regulator of DNA replication inhibited by PERK [47]. Finally, several highly upregulated proteins such as TRAIL, BAD and HTRA2 have been extensively studied as being apoptotic drivers in MM [48,49,50,51,52].

## 4. Materials and Methods

### 4.1. CD138^+^ Plasma Cell and Human MM Cell Lines

Bone marrow aspirates from 25 MM patients at diagnosis were collected after a written consent form was completed for the collection and analysis of samples according to the declaration of Helsinki. This research was approved by the Institutional Review Board (IRB)/Scientific committee of “Alexandra” Hospital on 18/10/2017. In order to separate mononuclear cells from the whole bone marrow aspirate, the samples were centrifuged by gradient centrifugation on Ficoll-Hypaque medium (Biochrom, Berlin, Germany). Cells were purified using CD138 microbeads according to the manufacturer’s instructions (Miltenyi Biotech, BergischGladbach, Germany).

MM cell lines U266 (TIB 196), H929 (CRL-9068) and RPMI-8226 (CCL-155) were purchased from the American Type Culture Collection (ATCC, Manassas, VA, USA); JJN3and L363 were kindly provided by Dr. C. Mitsiades (Dana-Farber Cancer Institute, Boston, USA); JIM3, KMS11 and OPM2 were kindly provided by Prof. A. Karadimitris (Imperial College London, London, UK); the ES2 ovarian cancer cell line was a kind gift from Prof. A. Scorilas (National and Kapodistrian University of Athens, Athens, Greece). MM cell lines were cultured in RPMI1640 GlutaMAX-I supplemented with 10% heat-inactivated fetal bovine serum (Invitrogen, Paisley, UK) and 1% penicillin/streptomycin (Biosera, Boussens, France). The ES2 cell line was cultured in McCoy’s 5a medium (Invitrogen, Paisley, UK) supplemented with 10% heat-inactivated fetal bovine serum and 1% penicillin/streptomycin. Cultures were kept at 37 °C in a humidified atmosphere of 5% carbon dioxide.

### 4.2. Reagents

Cells were treated with Tunicamycin (Sigma-Aldrich, Taufkirchen, Germany), a reagent that inhibits N-linked glycosylation, leading to the accumulation of misfolded proteins. To inhibit PERK activity, cells were treated with GSK2606414 (GlaxoSmithKline, Middlesex, UK). GSK2606414 was dissolved in dimethylsulfoxide (DMSO) to a stock concentration of 10 mM and stored in the dark at −20 °C. Commercially available BTZ was used.

### 4.3. Cell Survival/Viability Assay

The in vitro cell survival assay Water Soluble Tetrazolium Salt-1 (cleavage of the tetrazolium salt WST-1 (4-[3-(4-iodophenyl)-2-(4-nitrophenyl)-2H-5-tetrazolio]-1,3-benzenedisulfonate to formazan) was performed according to the manufacturer’s instructions (Clontech, Mountainview, USA). Cells were plated at 10^6^ cells/mL and incubated for 24–96 h at 37 °C. Plates were read at 450 nm wavelength on a DAS plate reader (Rome, Italy).

### 4.4. Apoptosis Assay

To determine cell apoptosis, myeloma cell lines treated with the PERK inhibitor were stained with Annexin V–fluoresceinisothiocyanate (Annexin V–FITC) and PI (FITC Annexin V apoptosis detection kit with PI, Biolegend, Germany) according to the manufacturer’s instructions. Stained cells (1 × 10^6^ per reaction) were analyzed on a flow cytometer (FACSCanto II, BD Biosciences, Germany). The Trypan blue exclusion method was also used to determine cell death. Apoptosis was further determined by the proteome profiler human apoptosis array kit (ARY009), a membrane-based antibody array, which detects 35 human apoptosis-related proteins simultaneously. To determine the average signal, the apoptosis array included a pair of duplicate spots representing each apoptosis-related protein.

### 4.5. RNA Extraction, Quantification and Amplification

RNA extraction was performed using the RNA extraction kit (Macherey-Nagel, Düren, Germany) according to the manufacturers’ instructions. Following RNA extraction, samples were quantified using the Qubit spectrophotometer (Invitrogen, Paisley, UK). Equal amounts of RNA were used for synthesis using the PrimeScript 1st strand cDNA synthesis kit (Takara, Shiga, Japan).

### 4.6. Real-Time Polymerase Chain Reaction

To perform quantitative Real-Time PCR (Q-RT-PCR), cDNA was processed using the SYBR Green Real-Time polymerase chain reaction (KapaBiosystems, Wilmington, MA, USA) technology according to the manufacturers’ instructions. Data were analyzed using the StepOnePlus Real time PCR system (Thermo Scientific, Waltham, MA, USA). Human *β-Actin* was set as the reference gene. Relative quantification was performed using 2^-(ΔΔCt)^. Untreated samples were used as calibrators. Primer sequences for Q-RT-PCR were designed using the Primer3 software and were as follows: *PERK*, Forward 5′-ATTGCATCTGCCTGGTTAC-3′, Reverse 3′-ACAGGCAAAGGAAGGAGTC-5′; *CHOP*, Forward 5′-TGGAAATGAAGAGGAAGAATCAAAAA-3′, Reverse 3′-CAGCCAAGCCAGAGAAGCA-5′; *ATF4*, Forward 5′-TGAAGGAGTTCGACTTGGATGCC-3′, Reverse 3′-GAAACCATGCCAGATGACCTTCTG-5′; *β-ACTIN*, Forward 5′-CCCTGGCACCCAGCAC-3′, Reverse 3′-GCCGATCCACACGGAGTAC-5′.

### 4.7. Polymerase Chain Reaction (PCR)

PCR was mainly performed to verify our results from RT-PCR. For the 50 µL reaction, we used platinum *Taq* reaction containing 4 pmol of the above primers (Invitrogen) with 1 Unit of platinum *Taq* DNA polymerase and 200 µM dNTPs, as described in the manufacturer’s instructions. The thermal conditions were 95 °C for 2 min, followed by 30 cycles of (94 °C for 15 s and 60 °C for 1 min), and finally 72 °C for 30 s. PCR products were analyzed using 2% ethidium bromide agarose gels.

### 4.8. siRNA Transfection

For RNAi analyses, L363 and H929 cells seeded in 6-well plates were transfected by using DharmaFECT Transfection reagent and either the SMARTpoolAccell EIF2AK3 (PERK) siRNA (E-004883-00-0005) or the Accell non-targeting pool (si-CTRL) (D-001910-01-05) (GE Healthcare Dharmacon Inc., Lafavette, CO, USA) according to the manufacturer’s instructions.

### 4.9. Immunoblotting Analysis

Before and after appropriate treatments, cells were spun at 1000 rpm for 5 min and pellets were washed with ice-cold phosphate-buffered saline (PBS). After PBS washes, pellets were resuspended with 100 μL per 1 × 10^6^ cells NP-40 buffer supplemented with protease/phosphatase inhibitor cocktail (Pierce, Rockford, IL, USA) and left on ice for 30 min. Following the last incubation, samples were spun at 3500 rpm for 15 min at 4 °C and protein lysates were collected and stored at −80 °C. Lysates were quantified by Bradford assay (Bio-Rad, Hertfordshire, UK) and equal amounts of protein were loaded on sodium-dodecyl sulfate–polyacrylamide gel electrophoresis (SDS–PAGE) for separation. After separation, samples were transferred to polyvinylidenedifluoride (PVDF) membranes followed by blockage with 5% non-fat milk and overnight incubation at 4 °C with the primary antibody and subsequent 1 h incubation with the appropriate secondary antibody at room temperature. The primary antibodies used were as follows: CHOP (Cell Signaling Technology, cat no: 2895), eIF2α (Cell Signaling Technology, cat no: 9722), p-eIF2α (Cell Signaling Technology, cat no: 9721), ATF4 (Cell Signaling Technology, cat no: 11815), PERK (Cell Signaling Technology, cat no: 3192), β-Actin (Cell Signaling Technology, cat no: 4967), ATF6 (Cell Signaling Technology, cat no: 65880) and XBP1s (Cell Signaling Technology, cat no: 12782S). Secondary antibodies conjugated to horseradish peroxidase were also obtained from Cell Signaling Technology. Detection was achieved by ECL-Plus (Amersham Biosciences, Little Chalfont, UK).

### 4.10. UPR Gene Expression Profiling

H929 cells were treated with DMSO or 10 μM GSK2606414 for 1 h. Cells were harvested and RNA isolation using the QiagenRNeasy spin column method was performed, followed by cDNA synthesis, as described above. Real-time PCR analysis, using a human UPR PCR array developed by Qiagen, was assayed on the LightCycler480 (Roche, Basel, Switzerland). Data were analyzed by the RT2 Profiler PCR Array data analysis software, provided by SABiosciences. Fold change in expression was determined using the ΔΔ*C*t method. Values used in downstream analysis were derived by taking the mean of the fold change of three biological replicates.

### 4.11. Statistical Analysis

Experiments were performed in duplicate or triplicate (for each biological replicate, *n* ≥ 2). Data points presented in graphs correspond to the mean of independent experiments. Student’s *t*-test was used for the determination of statistical significance. Data are presented as the mean ± standard deviation (SD). Significance at *p* < 0.05 is indicated by one asterisk, while significance at *p* < 0.01 is indicated by two asterisks. Microsoft Excel and GraphPad Prism 5.0 software were used for statistical analysis.

## 5. Conclusions

Taken together, our data show that the addiction of myeloma cells to high expression levels of UPR may be a key point in myeloma treatment and hence deserves further focused investigation. Our data also support the significance of the PERK pathway in MM cell biology and its inhibition as a potential anti-myeloma strategy. Given the on-target pharmacological effects of the PERK inhibitor GSK2606414 on MM cells, the development of even more potent PERK inhibitors may offer a therapeutic advantage (likely in combination with proteasome inhibitors) that would affect MM pathogenesis and treatment.

## Figures and Tables

**Figure 1 cancers-12-02864-f001:**
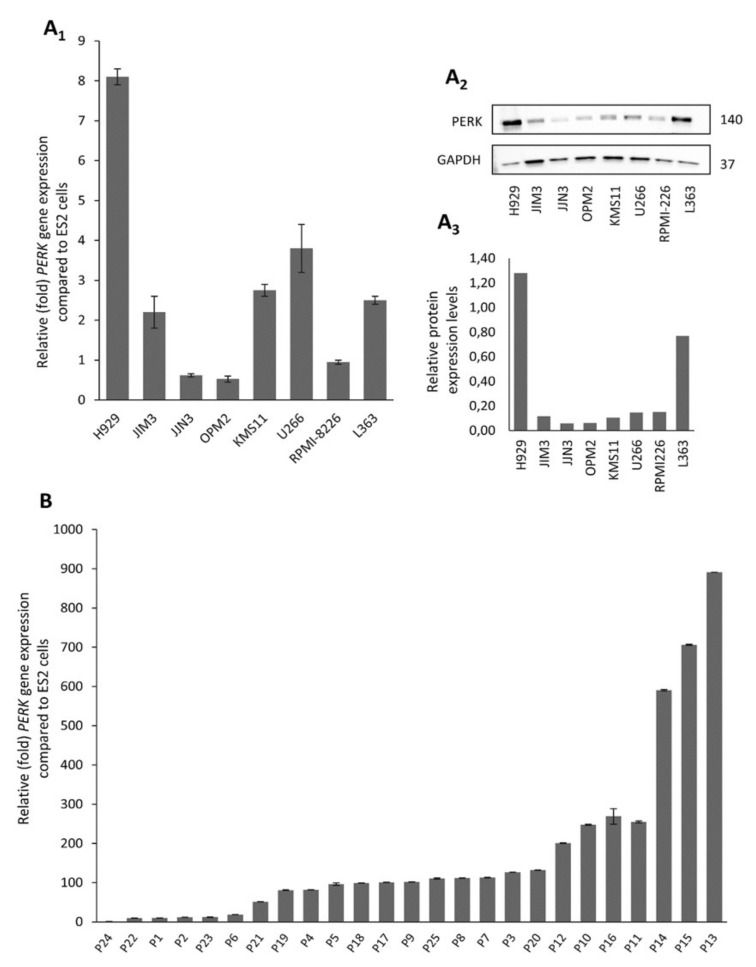
Protein kinase R (PKR)-like ER kinase (*PERK*) mRNA (**A_1_**) and protein (**A_2_**,**A_3_**) expression levels in multiple myeloma (MM) cell lines; the uncropped Western Blot figure is shown in Appendix A. (**B**) *PERK* mRNA expression in isolated CD138^+^ cells from selected MM patients (*n* = 25), as determined by Q-RT-PCR. Probing with glyceraldehyde 3-phosphate dehydrogenase (GAPDH) was used as total protein loading reference, whereas *β-ACTIN* gene expression was used as reference for RNA input. In graphs, means ± SDs from two replicates are shown.

**Figure 2 cancers-12-02864-f002:**
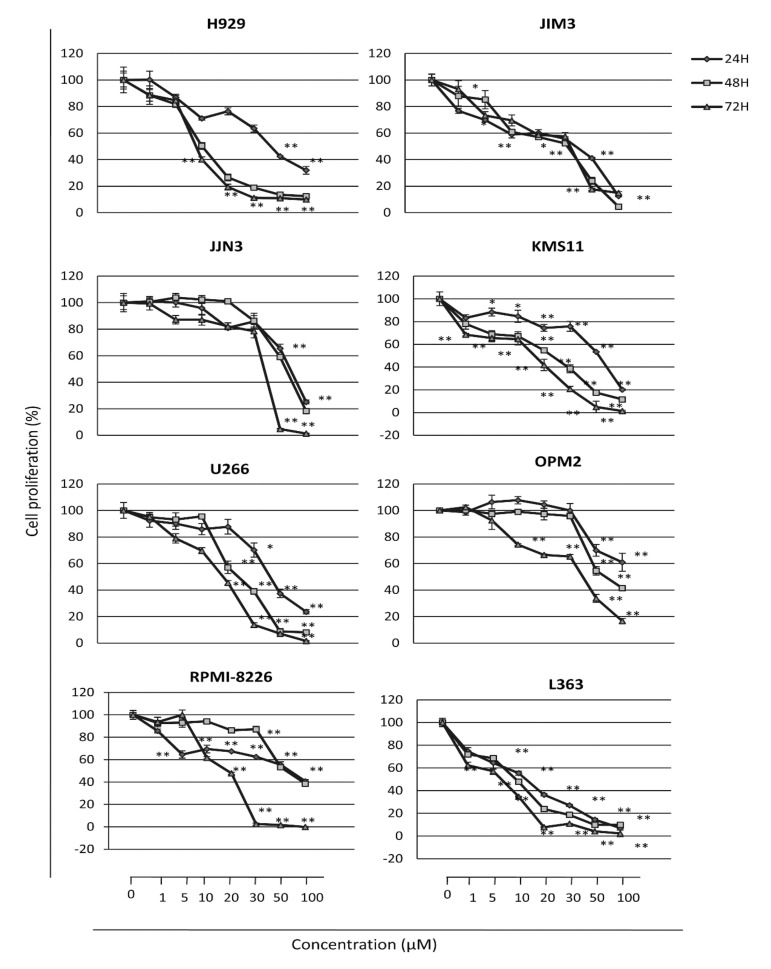
Cell survival of MM cell lines exposed to increasing doses of GSK2606414 for 24, 48 and 72 h, as determined by the Water Soluble Tetrazolium Salt 1 (WST-1) assay. Means ± SDs from three replicates are shown. *, *p* < 0.05; **, *p* < 0.01.

**Figure 3 cancers-12-02864-f003:**
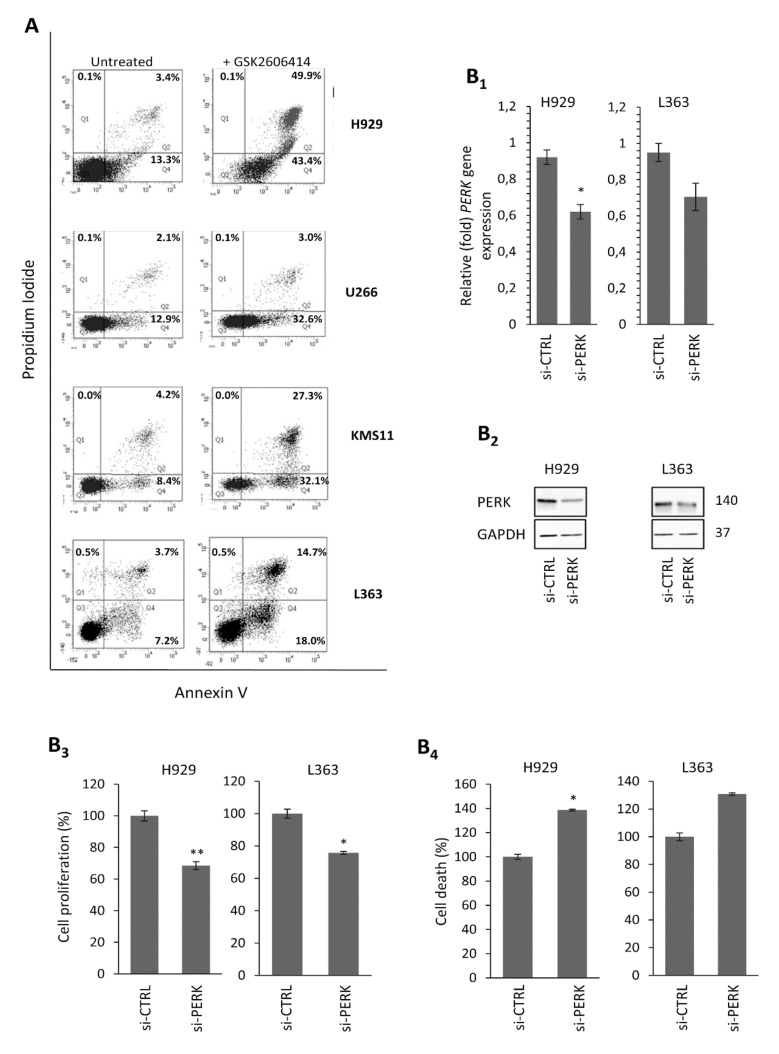
(**A**) Annexin V–propidium iodide (PI) staining of MM cell lines exposed to GSK2606414 for 48 h; to setup compensation and quadrants, unstained cells, cells stained only with Annexin V and cells stained only with PI were used as controls. Representative dot plots are shown out of two experiments performed. (**B_1_**) Q-PCR assay of *PERK* mRNA expression levels after transfecting H929 and L363 cell lines with *PERK* RNAi oligonucleotides or a non-targeting pool (si-CTRL) for 72 h. (**B_2_**) Immunoblotting analysis of PERK protein expression levels after transfecting H929 and L363 cell lines with *PERK* RNAi oligonucleotides or a non-targeting pool (si-CTRL) for 72 h. (**B_3_**) % Cell survival and (**B_4_**) % cell death in H929 and L363 cells after *PERK* RNAi inhibition for 72 h. GAPDH probing and *β-ACTIN* mRNA expression were used as references for total protein and mRNA input, respectively. Numbers next to bands refer to the molecular weight of proteins (MWs). The uncropped Western Blot figure is shown in Appendix A. Means ± SDs from three replicates are shown. *, *p* < 0.05; **, *p* < 0.01.

**Figure 4 cancers-12-02864-f004:**
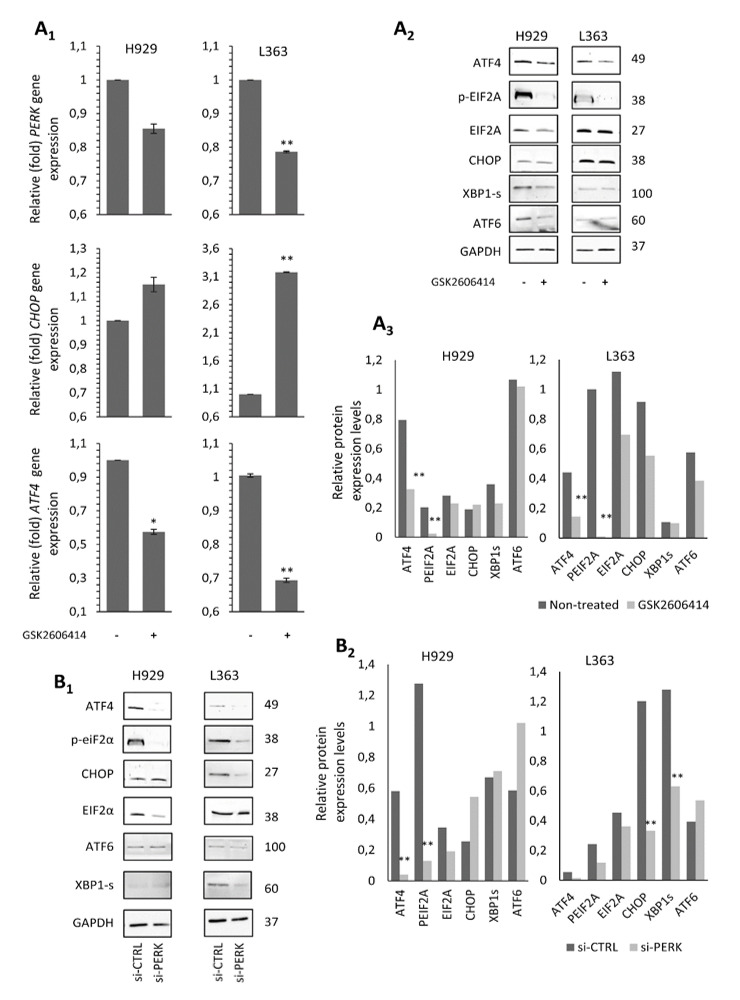
(**A_1_**) Q-PCR expression analyses of *PERK*, *ATF4* and *CHOP* mRNA expression levels in the H929 and L363 MM cell lines before (−) and after (+) incubation with 10 μM GSK2606414 for 48 h. (**A_2_**) Representative immunoblotting analyses of protein samples probed with antibodies against ATF4, p-eIF2A, eIF2A, CHOP, XBP1s and ATF6. (**A_3_**) Quantitation of expression levels of analyzed proteins. (**B_1_**) Immunoblotting analysis of protein samples probed with antibodies against ATF4, p-eIF2A, eIF2A, CHOP, ATF6 and XBP1s after transfecting H929 and L363 cell lines with *PERK* RNAi oligonucleotides or a non-targeting pool (si-CTRL) for 72 h. (**B_2_**) Quantitative analysis of protein expression levels in the immunoblots shown in (**B_1_**). GAPDH probing and *β-ACTIN* mRNA expression were used as references for total protein and mRNA input, respectively. Numbers next to bands refer to MWs. The uncropped Western Blot figure is shown in Appendix A. Means ± SDs from three replicates are shown. *, *p* < 0.05; **, *p* < 0.01.

**Figure 5 cancers-12-02864-f005:**
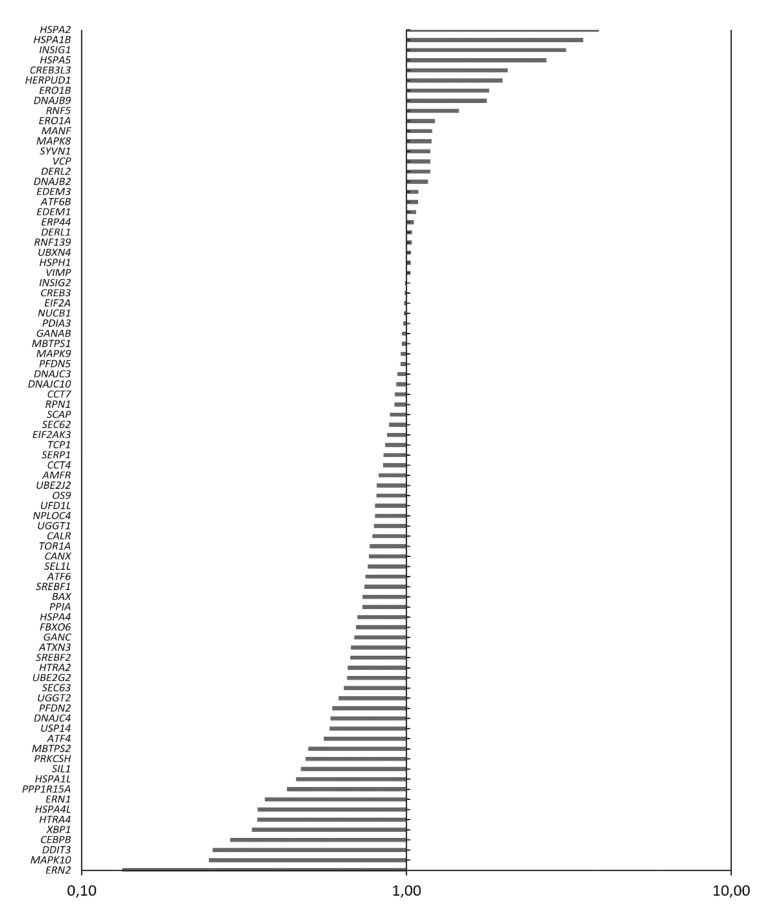
Q-PCR assay of 84 UPR-related genes following exposure of the H929 cell line to 10 μM GSK2606414 for 24 h and further subjected to Tunicamycin-induced endoplasmic reticulum (ER) stress for another 24 h. Results for GSK2606414-treated vs. untreated cells are shown.

**Figure 6 cancers-12-02864-f006:**
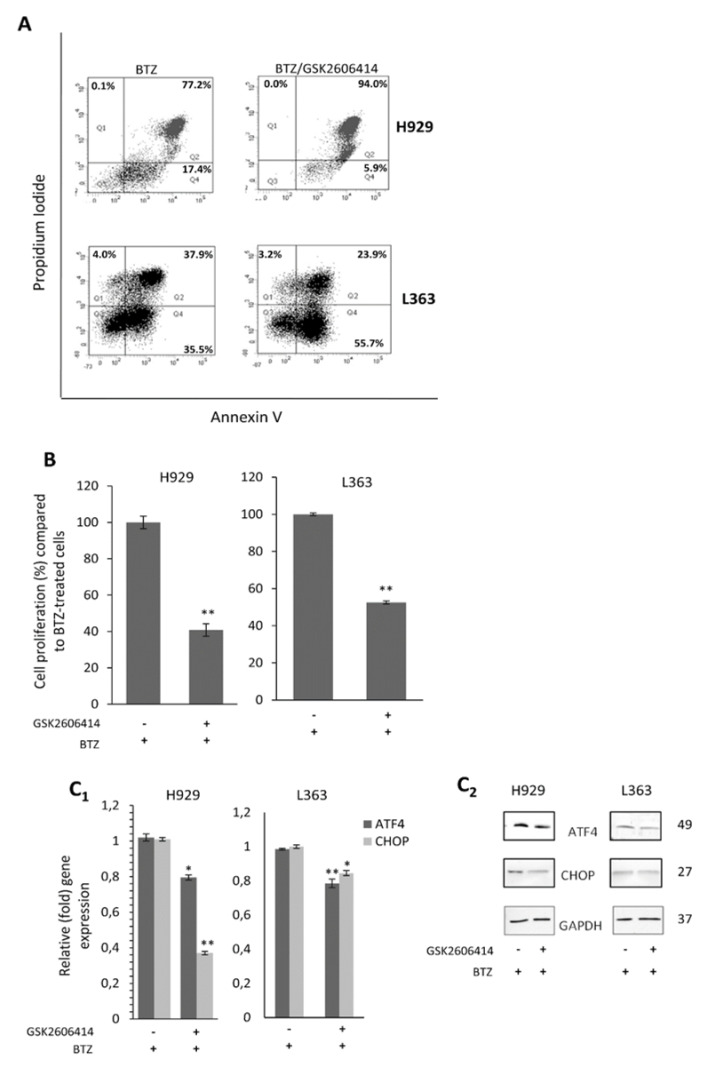
(**A**) Annexin V–PI staining of H929 and L363 cells exposed to bortezomib (BTZ; 4 nM) for 24 h with or without GSK2606414 (10 μM) treatment for 48 h; to set up compensation and quadrants, unstained cells, cells stained only with Annexin V and cells stained only with PI were used as controls. (**B**) % Cell survival of H929 and L363 cells subjected to BTZ (4 nM) for 24 h with or without GSK2606414 (10 μM) treatment for 48 h. (**C_1_**) Q-PCR expression analyses of *ATF4* and *CHOP* mRNA expression levels in H929 and L363 cells before and after incubation with BTZ (4 nM) for 24 h in the presence (or not) of GSK2606414 (10 μM) for 48 h. (**C_2_**) Representative immunoblotting analyses of protein samples probed with antibodies against ATF4 and CHOP. GAPDH probing and *β-ACTIN* mRNA expression were used as references for total protein and mRNA input, respectively. The uncropped Western Blot figure is shown in Appendix A. Means ± SDs from three replicates are shown. *, *p* < 0.05; **, *p* < 0.01.

**Figure 7 cancers-12-02864-f007:**
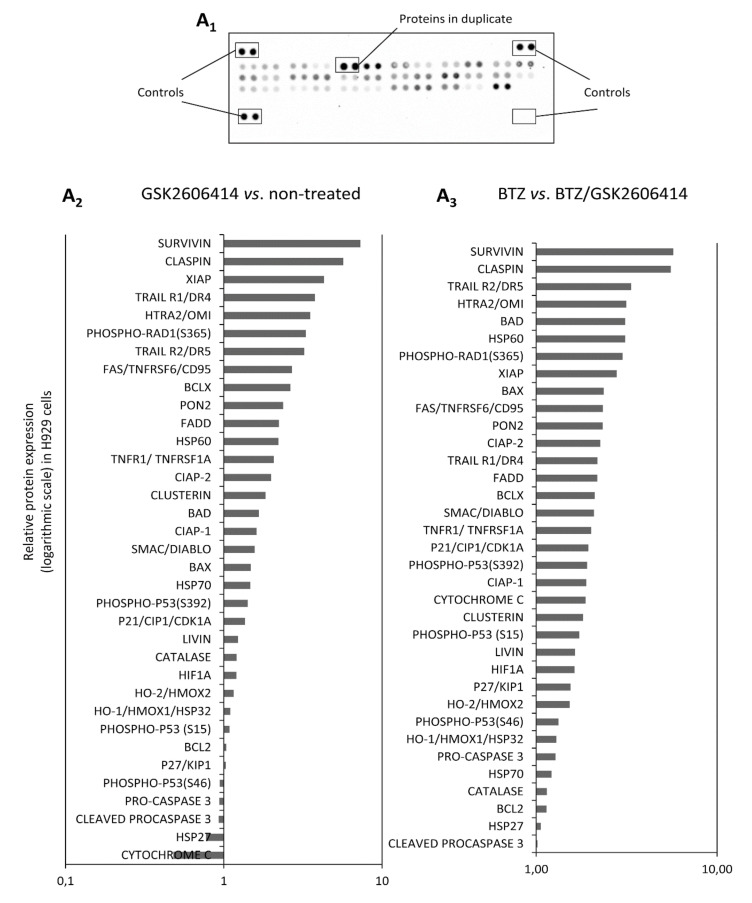
(**A_1_**) Apoptosis related proteome profiler. (**A_2_**) Relative protein expression of the shown 35 apoptosis-related proteins in GSK2606414 (10 μM)-treated vs. non-treated H929 cells. (**A_3_**) Relative protein expression of the 35 apoptosis-related proteins in BTZ and GSK2606414 (10 μM)-treated vs. BTZ (4 nM)-treated H929 cells.

**Table 1 cancers-12-02864-t001:** IC_50_ values of GSK2606414-treated cells at 48 h post-treatment.

Cell Lines	IC_50_ Values (μM) *
H929	10 ± 0.04
JIM3	29± 0.01
JJN3	75± 0.2
KMS11	22± 0.035
L363	9.5± 0.251
OPM2	75± 0.641
RPMI-8226	61± 0.56
U266	21± 0.08

* For calculating IC_50_, the four-parameter logistic function was used.

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
