# Peer review of "Characterization of a PERK Kinase Inhibitor with Anti-Myeloma Activity"

_cancers, 2020, doi:10.3390/cancers12102864_

Round 1

Reviewer 1 Report

Bagratuni et al. showed that both GSK2606414 treatment and PERK knockdown reduce the proliferation of malignant plasma cells and increase apoptosis of malignant plasma cells. They also showed that GSK2606414 combination with bortezomib exerted an additive toxic effect in malignant plasma cells. Although the topic of this manuscript is potential interesting, the role of PERK in tumor biology has been studies extensively. Moreover, this study fall short of exploring the mechanism by which PERK inhibition alters the proliferation and apoptosis of malignant plasma cells. There are a number of major concerns.

  1. The role of PERK in tumor cell proliferation and apoptosis has been studies extensively. Importantly, it has been show that PERK activation can promote or inhibit proliferation or apoptosis of tumor cells ( Ranganathan et al., Cancer Res. 2008 May 1;68(9):3260-8; Stone et al., Oncotarget. 2016 Sep 27;7(39):64124-64135; Ho et al., Am J Pathol. 2016 Jul;186(7):1939-1951; among others). The authors should cite and discuss these previous studies.
  2. WST-1 assay is not a cell proliferation assay. To determine cell proliferation, the authors should perform BrdU labeling or other similar assays.
  3. Fig 4, western blot should be quantified. The graphs do not have error bars.
  4. Fig 5, it is interesting that PERK inhibition alters the expression of a number of genes. The authors should determine the mechanism by which PERK inhibition alters the expression of these genes. The authors should also discuss how alternation of these gene expression contributes to the effects of PERK inhibition on malignant plasma cells.
  5. The major findings from the UPR gene expression profiling should be verified by real-time PCR.
  6. Fig 7, it is interesting that PERK inhibition alters the levels of a number of proteins. The authors should determine the mechanism by which PERK inhibition alters the levels of these proteins. The authors should also discuss how alternation of these protein levels contributes to the effects of PERK inhibition on malignant plasma cells.
  7. The major findings from dot blot should be verified by western blot.

Author Response

Bagratuni et al. showed that both GSK2606414 treatment and PERK knockdown reduce the proliferation of malignant plasma cells and increase apoptosis of malignant plasma cells. They also showed that GSK2606414 combination with bortezomib exerted an additive toxic effect in malignant plasma cells. Although the topic of this manuscript is potential interesting, the role of PERK in tumor biology has been studies extensively. Moreover, this study fall short of exploring the mechanism by which PERK inhibition alters the proliferation and apoptosis of malignant plasma cells. There are a number of major concerns

1.         The role of PERK in tumor cell proliferation and apoptosis has been studies extensively. Importantly, it has been show that PERK activation can promote or inhibit proliferation or apoptosis of tumor cells ( Ranganathan et al., Cancer Res. 2008 May 1;68(9):3260-8; Stone et al., Oncotarget. 2016 Sep 27;7(39):64124-64135; Ho et al., Am J Pathol. 2016 Jul;186(7):1939-1951; among others). The authors should cite and discuss these previous studies. 

OUR RESPONSE AND REVISIONS:

As suggested by the Reviewer, we have now described in the introduction section findings from other studies. The inserted text is shown by the track-changes tool.

  1. WST-1 assay is not a cell proliferation assay. To determine cell proliferation, the authors should perform BrdU labeling or other similar assays.

OUR RESPONSE AND REVISIONS:

We would like to thank the Reviewer for this valid comment. We absolutely agree that WST1 is rather a cell survival assay. The reason that we refer to it as a proliferation assay is due to the fact that previous reviewers have insisted that it is rather a proliferation assay. We have now changed the statement “cell proliferation assay” to “survival/viability assay” in the entire manuscript.

  1. Fig 4, western blot should be quantified. The graphs do not have error bars.

OUR RESPONSE AND REVISIONS:

We would like to thank the Reviewer for the comment. In Fig. 4 the western blots are quantified. Specifically for western blot in Fig 4A2 the quantification is shown in Fig 4A3, while for western blot in Fig 4B1 the quantification is shown in Fig 4B2. However errors bars in Fig 4A3 and 4B2 could not be included as the quantification is derived from the single western blot presented in Fig 4A2 and Fig 4A1 respectively, and not from the duplicate/experiments we have conducted.

  1. Fig 5, it is interesting that PERK inhibition alters the expression of a number of genes. The authors should determine the mechanism by which PERK inhibition alters the expression of these genes. The authors should also discuss how alternation of these gene expression contributes to the effects of PERK inhibition on malignant plasma cells.

OUR RESPONSE AND REVISIONS:

As suggested by the Reviewer, we have now added in the discussion section the information regarding the differential expression of the genes from the UPR profile assay and how this might contribute to MM cell survival. Please refer to lines 237-252 of the discussion section.

  1. The major findings from the UPR gene expression profiling should be verified by real-time PCR.

OUR RESPONSE AND REVISIONS:

We would like to thank the Reviewer for the insightful observation. The main focus of this study is to determine the effects of PERK inhibition in the survival of MM cells based on the PERK- ATF4 pathway of the UPR, which is a highly activated branch of the UPR in MM. Therefore, our main analysis is to extensively study the genes involved in this pathway which is also the main target of the GSK2606414 inhibitor. Using the panel of 84 UPR gene expression profile, we aim to reveal other pathways or transcription factors which might be affected by PERK inhibition as a general observation as to what other potential key players might contribute to MM biology through the UPR system that could affect proliferation, apoptosis or resistance to therapy. Therefore, at this stage we could not further proceed to the verification and or analysis of the major findings from the UPR gene expression profiling.

  1. Fig 7, it is interesting that PERK inhibition alters the levels of a number of proteins. The authors should determine the mechanism by which PERK inhibition alters the levels of these proteins. The authors should also discuss how alternation of these protein levels contributes to the effects of PERK inhibition on malignant plasma cells.

OUR RESPONSE AND REVISIONS:

As suggested by the Reviewer, we have now added in the discussion section the information regarding the differential expression of proteins and how these alterations might contribute to malignant plasma cell biology. Please refer to lines 265-272 of the discussion section.

  1. The major findings from dot blot should be verified by western blot.

OUR RESPONSE AND REVISIONS:

We would like to thank the Reviewer for the insightful observation. As previously mentioned, the main focus in this study is to determine the effects of PERK inhibition in the survival/apoptosis of MM cells based on the PERK- ATF4 pathway of the UPR, which is a highly activated branch of the UPR in MM. Therefore, our main analysis is to extensively study the proteins involved in this pathway which is also the main target of the GSK2606414 inhibitor as well as the apoptosis which occurs through the inhibition of this pathway. Using the apoptotic array, we aim to further understanding how the apoptosis phenotype seen by Annexin/ PI flow cytometry is coupled with the expression of a panel apoptotic proteins and how these proteins could have an effect in the behavior of malignant plasma cells. Therefore, at this stage we could not further proceed to the verification and or analysis of the major findings from the human profiler apoptotic assay

Reviewer 2 Report

Very interesting paper that I can recommend for publication.

I have minor issues author perhaps can improve.

line 49: Minor spelling error

All figures: Are there no figure text?

Figure 1A1: Why was the ES2 cell line chosen for control, explain. How is ES2 PERK expression compared to e.g. healthy plasma cells

Figure 1B: How many patients had an expression below the ES2 cell line

Section 2.2 well written

Figure 3: Figure text? or explain why propidium iodide on the x axis. Explain CTRL (control?)

Section 2.3 and 2.4 Why was H929 and L363 chosen for further experiments. Cell line KMS11 and U266 had greater PERK gene expression than L363?

Section 2.4 line 147: Does the author have any idea why they saw a different CHOP level in H929 and L363, but in both cell lines saw a clear anti proliferative effect of SK2606414 

Author Response

Very interesting paper that I can recommend for publication.

I have minor issues author perhaps can improve.

  1. line 49: Minor spelling error

OUR RESPONSE AND REVISIONS:

“Line 49: checkpoint for plasma cell survival. Multiple myeloma (MM) plasma cells usually produce large”. We would like to kindly reply that we are not sure where is the spelling error. If you could please specify, we will make the correction.

  1. All figures: Are there no figure text?

OUR RESPONSE AND REVISIONS:

Admittedly, the comment of the Reviewer is valid. Although we submitted the figure legends, they are not included in the manuscript format. We will include them in the revised manuscript.

  1. Figure 1A1: Why was the ES2 cell line chosen for control, explain. How is ES2 PERK expression compared to e.g. healthy plasma cells

OUR RESPONSE AND REVISIONS:

We would like to kindly respond that the ovarian cancer cell line ES2 was chosen as we have previously shown that the UPR is not activated in these cells [1], hence we believe is a good non B cell line control. However, we are working on obtaining healthy plasma cells in the near future, as we understand how important is to include them as controls for these type of studies.

  1. Figure 1B: How many patients had an expression below the ES2 cell line

OUR RESPONSE AND REVISIONS:

We would like to kindly reply that no MM patient had a PERK expression lower than ES2 cell line.

  1. Section 2.2 well written

OUR RESPONSE AND REVISIONS:

We would like to thank the Reviewer for the comment.

  1. Figure 3: Figure text? or explain why propidium iodide on the x axis. Explain CTRL (control?)

OUR RESPONSE AND REVISIONS:

We would like to thank the Reviewer for the comment. As mentioned before, the figure legends will be included in the revised manuscript. As far as we can see in Fig 3, propidium iodide is on y axis. However, if something does not seem appropriate, we are willing to correct it. CTRL is control (non targeting pool) and has been described in Figure legends.

  1. Section 2.3 and 2.4 Why was H929 and L363 chosen for further experiments. Cell line KMS11 and U266 had greater PERK gene expression than L363?

OUR RESPONSE AND REVISIONS:

We would like to thank the Reviewer for the comment. These experiments were done in H929 and L363 cells, because they express high endogenous PERK levels but most importantly they are highly responsive to GSK2606414-induced PERK inhibition as seen in Fig 3A.

  1. Section 2.4 line 147: Does the author have any idea why they saw a different CHOP level in H929 and L363, but in both cell lines saw a clear anti proliferative effect of GSK2606414 

OUR RESPONSE AND REVISIONS:

We would like to thank the Reviewer for the insightful observation. Indeed treatment with GSK260641 had a different effect on CHOP expression in the two cell lines. Although we do not have a definite answer to this question, we believe that possibly PERK knockdown in H929 cells is not enough in order to produce downstream effects leading to CHOP suppression.  

  1. Bagratuni, T., A.D. Sklirou, and E. Kastritis, Toll-Like Receptor 4 Activation Promotes Multiple Myeloma Cell Growth and Survival Via Suppression of The Endoplasmic Reticulum Stress Factor Chop. Sci Rep 2019. 9(1): p. 3245.